# Learning Versatile Filters for
# Efficient Convolutional Neural Networks

**Yunhe Wang[1], Chang Xu[2], Chunjing Xu[1], Chao Xu[3], Dacheng Tao[2]**
[1] Huawei Noah's Ark Lab
[2] UBTECH Sydney AI Centre, SIT, FEIT, University of Sydney, Australia
[3] Key Lab of Machine Perception (MOE), Cooperative Medianet Innovation Center,
School of EECS, Peking University, Beijing, China
`yunhe.wang@huawei.com, c.xu@sydney.edu.au, xuchunjing@huawei.com`
`xuchao@cis.pku.edu.cn, dacheng.tao@sydney.edu.au`

## Abstract

This paper introduces versatile filters to construct efficient convolutional neural network. Considering the demands of efficient deep learning techniques running on cost-effective hardware, a number of methods have been developed to learn compact neural networks. Most of these works aim to slim down filters in different ways, *e.g.* investigating small, sparse or binarized filters. In contrast, we treat filters from an additive perspective. A series of secondary filters can be derived from a primary filter. These secondary filters all inherit in the primary filter without occupying more storage, but once been unfolded in computation they could significantly enhance the capability of the filter by integrating information extracted from different receptive fields. Besides spatial versatile filters, we additionally investigate versatile filters from the channel perspective. The new techniques are general to upgrade filters in existing CNNs. Experimental results on benchmark datasets and neural networks demonstrate that CNNs constructed with our versatile filters are able to achieve comparable accuracy as that of original filters, but require less memory and FLOPs.

## 1 Introduction

Considerable computer vision applications (*e.g.* image classification [19], object detection [15], subspace clustering [27], and image segmentation [13]) have received remarkable progress with the help of convolutional neural networks (CNNs) in last decade. Table 1 summarizes profiles of benchmark CNNs on the ILSVRC 2012 dataset [17]. From the pioneering AlexNet [11] to the recent ResNeXt-50 [25], the storage of networks is slightly saved, but the classification accuracy has been continuously improved. This performance improvement comes from sophisticatedly designed calculations introduced in these networks, *e.g.* residual modules in ResNet [7] and versatile modules in GoogleNet [20]. These networks are widely used in the scenario of abundant computation and storage resources, but they cannot easily adapt to mobile platforms, such as smartphones and cameras. Taking ResNet-50 [7] with 54 convolutional layers as an example, about $97MB$ memory is required to store all filters and over $4.0 \times 10^9$ times of floating number multiplications have to be operated for an image.

Over the years, different techniques have been proposed to tackle the contradiction between resources supply of low performance devices and demands of heavy neural networks. One common approach is to explore and eliminate redundancy in pre-trained CNNs. For example, Han *et.al.* [6] discarded subtle weights in convolution filters, Wang *et.al.* [23] investigated redundancy between weights, Figurnov *et.al.* [5] removed redundant connections between input data and filters, Wang *et.al.* [22]

Table 1: Properties of benchmark CNN models learned on the ILSVRC 2012 dataset.

| Model | Weights | Memory | FLOPs | Top1-err | Top5-err |
|---|---|---|---|---|---|
| AlexNet [11] | $6.1 \times 10^7$ | $232.5MB$ | $0.7 \times 10^9$ | $42.9\%$ | $19.8\%$ |
| VGGNet-16 [19] | $13.8 \times 10^7$ | $526.4MB$ | $15.4 \times 10^9$ | $28.5\%$ | $9.9\%$ |
| GoogleNet [20] | $0.7 \times 10^7$ | $26.3MB$ | $1.5 \times 10^9$ | $34.2\%$ | $12.9\%$ |
| ResNet-50 [7] | $2.6 \times 10^7$ | $97.2MB$ | $4.1 \times 10^9$ | $24.7\%$ | $7.8\%$ |
| ResNeXt-50 [25] | $2.5 \times 10^7$ | $95.3MB$ | $4.2 \times 10^9$ | $22.6\%$ | $6.5\%$ |

explored compact feature maps for deep neural networks, and Wen *et.al.* [24] investigated the sparsity from several aspects. There are also some methods to approximate the original neural networks by employing more compact structures, *e.g.* quantization and binarization [1, 14, 3], matrix decomposition [4], and teacher student learning paradigm [8, 16]. Instead of patching pre-trained CNNs, some highly efficient network architectures have been designed for applications on mobile devices. For example, ResNext [25] aggregated a set of transformations with the same topology, Xception [2] and MobileNet [9] used separable convolutions with $1 \times 1$ filters, and ShuffleNet [26] encouraged pointwise group convolutions and channel shuffle operations.

Most of these existing works learn efficient CNNs through slimming down filters, *e.g.* making great use of smaller filters (*e.g.* $1 \times 1$ filters) and developing various (*e.g.* sparse and low-rank) approximation of filters. Given such lightweight filters, the network performance is struggling to keep up, due to limited capacity of $1 \times 1$ filters or approximation error of filters. Rather then subtracting (*i.e.* slimming down filters), another thing to consider is adding. We must ask whether the value of a normal filter has already been maximally explored and can a normal filter take more roles than usual.

In this paper, we propose versatile filters for efficient convolutional neural networks. We produce a series of smaller secondary filters from a primary filter based on some pre-defined rules. These secondary filters inherit weights from the primary filter, but they will have different receptive fields and extract features from the spatial dimension. The neural network is composed of primary filters, while the strength of the network will be fully disclosed through secondary filters in computation. Specifically, we develop versatile filters in both spatial and channel dimensions. We provide detailed feed-forward and back-propagation of the proposed versatile filters. Experiments on benchmarks demonstrate that, equipping CNNs with our versatile filters can lead to lower memory usages and FLOPs, but with comparable network accuracy.

## 2 Approach

In this section, we illustrate the design of versatile filters, which can be applied over any filter with height and width greater than one. Besides spatial versatile filters, we additionally investigate versatile filters from the channel perspective.

### 2.1 Spatial Versatile Filters

Consider the input data $x \in \mathbb{R}^{H \times W \times c}$, where $H$ and $W$ are height and width of the input data respectively, and $c$ is the channel number, *i.e.* the number of feature maps generated in the previous layer. A convolution filter is denoted as $f \in \mathbb{R}^{d \times d \times c}$, where $d \times d$ is the size of the convolution filter. We focus on square filters, *e.g.* $5 \times 5$ and $3 \times 3$, which are most widely used in modern CNNs such as ResNet [7], VGGNet [19], ResNeXt [25], and ShuffleNet [26]. The conventional convolution can be formulated as

$$y = f * x, \tag{1}$$

where $*$ is the convolution operation, $y \in \mathbb{R}^{H' \times W'}$ is the output feature map of $x$, and $H'$ and $W'$ are its height and width, respectively.

Compared with traditional fully connected neural networks, one of the most important advantages of CNNs is that the size ($d \times d$) of filters in a convolutional layer can be much smaller than that ($H \times W$) of the input. For example, $7 \times 7$ filters in the first layer of ResNet-50 [7] are used to process the $224 \times 224$ input. Fixing the output size, the complexity of floating number multiplications of a filter in the fully-connected layer is $\mathcal{O}(cHWH'W')$, while the complexity of a convolution filter is

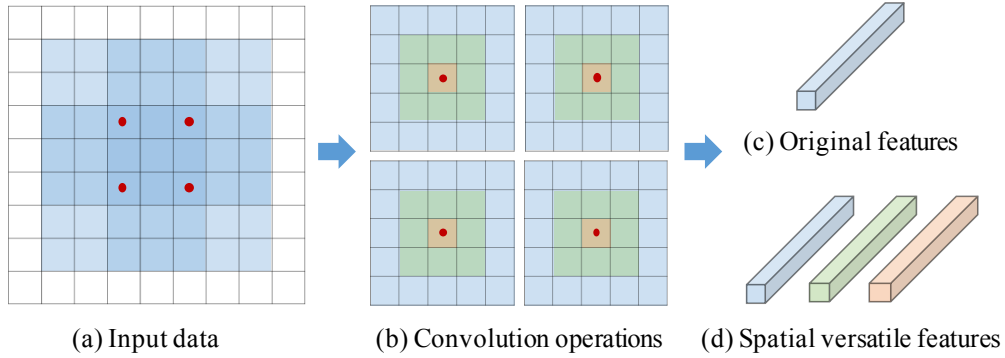

|     |     |     |
| :-: | :-: | :-: |
| (a) Input data | (b) Convolution operations | (d) Spatial versatile features |

Figure 1: An illustration of the proposed spatial versatile convolution filter. Given the input data (a), there are four sub-regions (b) covered by a $5 \times 5$ convolution filter with stride 2, and their convolution results are stacked into a feature map (c). In contrast, a spatial versatile filter will be applied three times on each sub-region with different secondary filters, *i.e.*, $5 \times 5$ blue, $3 \times 3$ green, and $1 \times 1$ red in (b) to generate three feature maps (d).

only $\mathcal{O}(cd^2 H'W')$. In addition, convolution operations extract features from small regions, which is beneficial for subsequent tasks such as recognition and detection.

Receptive field is an important concept introduced by convolutions. Larger receptive field would allow neurons to detect changes over a wider area, but result in a less precise perception. On the other hand, smaller receptive field would enable neurons to detect fine details. It is therefore reasonable to integrate neurons of larger receptive fields and smaller receptive fields to extract comprehensive and accurate features. For example, versatile modules [20] introduce parallel paths with different receptive field sizes by making use of multiple filters with different sizes, e.g. $3 \times 3$ and $5 \times 5$ convolutions. Explicitly brining in filters of different sizes is a straightforward approach to process the input information in different scales, but the significant increase in storage of these filters could be a new challenge. Most importantly, though filters of different sizes in the same layer have different receptive fields, their receptive fields would have some overlap, which indicates the prospective connections between their corresponding filters.

Taking $f \in \mathbb{R}^{d \times d}$ as a primary filter, we propose to derive a series of secondary filters $\{f_1, f_2, \cdots, f_s\}$ from $f$, where $s = \lceil d/2 \rceil$. To maximally explore the potential of primary filter $f$, each secondary filters $f_i$ is directly inherited from $f$ with a mask $M_i$,

$$M_i(p, q, c) = \begin{cases} 1, & if \ q, p \geq i \mid p, q \leq d + 1 - i, \\ 0, & otherwise, \end{cases} \quad (2)$$

and $f_i$ is calculated as $f_i = M_i \circ f$, where $\circ$ is the element-wise multiplication. More specifically, $f_1$ is the filter $f$ itself, $f_2$ discards the outermost circle of parameters in $f$, and $f_s$ is the innermost circle of parameters in $f$ (*i.e.*, $f_s$ is a $1 \times 1$ filter given an odd $d$). Example secondary filters for a $5 \times 5$ filter can be seen in Figure 1 (b).

By concatenating convolution responses from these secondary filters, we get the feature map represented as

$$y = [(M_1 \circ f) * x + b_1, ..., (M_s \circ f) * x + b_s],$$
$$s.t. \ \ s = \lceil d/2 \rceil, \ \ \{M_i\}_{i=1}^s \in \{0, 1\}^{d \times d \times c}, \quad (3)$$

where $b_1, ..., b_s$ are bias parameters.

By embedding Fcn. 3 into conventional CNNs, we can obtain convolution responses simultaneously from $s$ secondary filters of different receptive fields. The number of the output channels of the proposed versatile filter is $s$ times more than that of the original filter, and feature maps of a convolutional layer using the proposed versatile filters contain features in different scales at the same time.

Note that convolution operations ($*$) in Fcn. 3 share the same stride and padding parameters for the following two reasons: 1) dimensionalities of feature maps generated by secondary filters with

different receptive fields have to be consistent for the subsequent calculation; 2) centers of these secondary filters are the same, and the $s$-dimensional feature is thus a multi-scale representation of a specific pixel at $x$. The schematic of the proposed versatile filters is shown in Fig. 1, and the detailed back-propagation procedure of the proposed spatial versatile convolution filters can be found in the supplementary materials.

**Discussion:** Besides the proposed method as shown in Fcn. 3, a naïve approach to aggregate features from multiple secondary filters can be

$$y = \sum_{i=1}^{s} (M_i \circ f) * x + b, \tag{4}$$
$$s.t. \quad s = \lceil d/2 \rceil, \quad \{M_i\}_{i=1}^{s} \in \{0, 1\}^{d \times d \times c},$$

which calculates the resulting feature map as a linear combination of features from different receptive fields. Since the convolution $*$ is exactly an linear operation, the sum of different convolution responses on the same input can be rewritten as the response of a combined convolution filter employed on this data, *i.e.*,

$$y = \sum_{i=1}^{s} (M_i \circ f) * x + b = [(\sum_{i=1}^{s} M_i) \circ f] * x + b. \tag{5}$$

Therefore, Fcn. 4 is equivalent to adding a fixed weight mask on conventional convolution filters, which cannot produce more meaningful calculations and informative features in practice. We will compare the performance of this naïve approach in experiments.

## 2.2 Analysis on Spatial Versatile Filters

Compared with original convolution filters, the proposed spatial versatile filters can provide more feature maps without increasing the number of filters. Therefore, we further analyze the memory usage and computation cost of neural networks using the proposed spatial versatile filters.

The proposed spatial versatile convolution operation as shown in Fcn. 3 can generate multiple feature maps using a fixed number of convolution filters. Thus the computational complexity and memory usage of CNNs for extracting the same amount of features can be reduced significantly as analyzed in Proposition 1.

**Proposition 1.** *Given a convolutional layer for extracting feature maps $y \in \mathbb{R}^{H' \times W' \times n}$ using the proposed spatial versatile filters (Fcn. 3), the space complexity of $d \times d$ filters with $c$ channels is $\mathcal{O}(d^2 cn/s)$ and the computational complexity is $\mathcal{O}(\sum_{i=1}^{s}(d - 2i + 2)^2 cH'Wn/s)$.*

*Proof.* For the desired feature map $y \in \mathbb{R}^{H' \times W' \times n}$, where $H'$ and $W'$ are height and width of $y$, respectively. Commonly, we need $n$ convolution filters $\{f_i\}_{i=1}^{n}$ of size $d \times d \times c$. The space complexity for storing these filters is $\mathcal{O}(d^2 cn)$, and the computational complexity for generating $y$ is $\mathcal{O}(d^c H'W'n)$.

In contrast, the proposed spatial versatile convolution operation can extract $s = \lceil d_i/2 \rceil$ sets of feature maps simultaneously. Thus, for generating $N$ feature maps, the space complexity for storing the proposed spatial versatile convolution filters is

$$\mathcal{O}(d^2 cn/s). \tag{6}$$

The computational complexity for generating feature maps using the proposed spatial versatile filters in different scales is various, which affects by the size of convolution filters, *i.e.*, the number of non-zero elements in each $M_i$. The number of non-zero elements in $M_i$ is $(d - 2i + 2)^2$ as shown in Fcn. 2, thus the computational complexity for the $i$-th scale is $\mathcal{O}((d-2i+2)^2 cH'W'n/s)$. Therefore, the computational complexity of the entire layer can be calculated as:

$$\mathcal{O}(\sum_{i=1}^{s}(d - 2i + 2)^2 cH'W'n/s), \tag{7}$$

which is definitely smaller than that $\mathcal{O}(d^2 cH'W'n)$ of the traditional convolution operation when $s > 2$. □

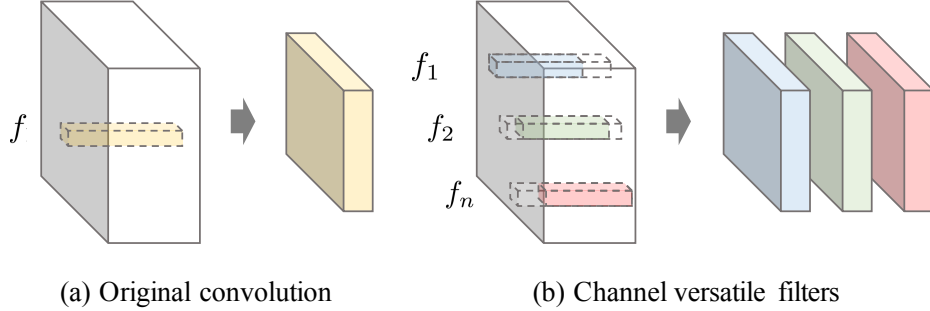

|  (a) Original convolution  |  (b) Channel versatile filters |

Figure 2: An illustration of the proposed channel versatile filters. The original filter can only generate only one feature map for the given input data, and the proposed method can provide multiple feature maps simultaneously according to the channel stride parameters. Each color represents a secondary filter and its corresponding feature map.

## 2.3 Channel Versatile Filters

A spatial versatile filter was proposed in Fcn. 3, which generates a series of secondary convolution filters by adjusting the height and width of a given convolution filter. However, there are still obvious redundancy in these secondary filters, $i.e.$, the number of channels of each convolution filter is much larger than its height and width. In addition, given $1 \times 1$ primary filters, Fcn. 3 will be reduced to Fcn. 1 for conventional convolution operation. Considering the wide use of $1 \times 1$ filters in modern CNN architectures such as ShuffleNet [26] and ResNeXt [25], $etc.$, we proceed to develop versatile filters from the channel perspective.

The most important property of convolution filters is that their weights are shared by the input data. A convolution filter used to have the same depth as the input data, and slide along the width and height of the input data with some stride parameters. If the depth of the input is 512, a $1 \times 1 \times 512$ filter has to take a large number of floating number multiplications to weight different channels and integrate the information across different input channels. However, this coarse information summarization over all channels is difficult to highlight characters of individual channels, especially when there are extremely many channels. Hence, we define secondary filters for original convolution filters with the help of channel stride, $i.e.$

$$
\begin{aligned}
y &= [f_1 * x + b_1, f_2 * x + b_2, \cdots, f_n * x + b_n], \\
&s.t. \ \forall \, i, \ f_i \in \mathbb{R}^{d \times d \times c}, \ n = (c - \hat{c})/g + 1.
\end{aligned}
\tag{8}
$$

where $g$ is the channel stride parameter and $\hat{c} < c$ is the number of non-zero channels of secondary filters. $f_i$ is the $i$-th unduplicated copy of primary filter $f$ given the length $\hat{c}$ and the stride $g$. Therefore, a filter will be used $n$ times simultaneously to generate more feature maps by introducing Fcn. 8. Example secondary filters using the proposed channel stride approach are given in Figure 2. In addition, the proposes channel versatile filters can also significantly reduce the memory usage and computational complexity of CNNs, which can be similarly derived as that in Proposition 1.

## 3 Experiments

In this section, we will implement experiments to validate the effectiveness of the proposed multi-scale convolution filter on several benchmark image datasets, including MNIST [12], ImageNet (ILSVRC 2012 [17]), $etc.$ Experimental results will be analyzed to further understand the benefits of the proposed approach.

### 3.1 Experiments on MNIST

The MNIST dataset consists of $70,000$ images drawn from ten categories, which is split into $60,000$ training and $10,000$ testing images. Each sample in this dataset is a $28 \times 28$ gray-scale digit (from 0 to 9) image. In addition, the last $10,000$ images in the training set is selected as the validation set for determining the final model.

**Spatial versatile filters:** We first tested the performance of the proposed spatial versatile filter in Fcn. 3 using a LeNet for classifying the MNIST dataset learned on MatConvNet [21]. The baseline model has four convolutional layers of size $5 \times 5 \times 1 \times 20$, $5 \times 5 \times 20 \times 50$, $4 \times 4 \times 50 \times 500$, and $1 \times 1 \times 500 \times 10$, respectively, which accounts about $1.6MB$ (filters are stored in 32-bit floating values), and the accuracy is $99.20\%$. Then, several models with different architectures and strategies were trained, and their results are show in Table 2. Wherein, memory usage of convolution filters and floating number multiplications (FLOPs) of each model are also provided.

Versatile-Model 1 is the network using the proposed versatile filters (Fcn. 4) with the same architecture as that of the baseline model. Since it does not change the size of output data, its memory usage and multiplications are also the same as those of the baseline model. Not surprisingly, there is not any performance enhancement by exploiting this approach since the network can adjust parameters in convolution filters according to the weight mask $\left(\sum_{i=1}^{s} M_i\right)$.

Table 2: The performance of the proposed spatial versatile filters on MNIST.

| Model | Weights | Memory | FLOPs | Accuracy |
|---|---|---|---|---|
| Baseline | $4.3 \times 10^5$ | $1681.6KB$ | $22.93 \times 10^5$ | $99.20\%$ |
| Versatile-Model 1 | $4.3 \times 10^5$ | $1681.6KB$ | $22.93 \times 10^5$ | $99.20\%$ |
| Versatile-Model 2 | $2.2 \times 10^5$ | $852.0KB$ | $12.00 \times 10^5$ | $99.15\%$ |
| Versatile-Model 3 | $2.2 \times 10^5$ | $852.0KB$ | $12.00 \times 10^5$ | $99.22\%$ |

Versatile-Model 2 and Versatile-Model 3 adopted the proposed versatile convolution operation as shown in Fcn. 3. There are multiple bias term $b_1, ..., b_s$ in Fcn. 3 for controlling features generated by different secondary filters from an versatile convolution filter. The difference between Model 2 and Model 3 is that, bias term of convolution filters in Model 3 are shared, *i.e.*, $b_1 =, ..., = b_s$, and gradients of $b$ are also averaged.

The proposed method can generate multiple feature maps using a convolution filter whose size is larger than $2 \times 2$ (*i.e.*, $s = \lceil d_i/2 \rceil > 1$), which will increase the number of channels in the next layer and make the convolutional neural network enormous. Therefore, we reduce the number of convolution filters in each layer to make the amount of feature maps in Versatile-Model 2 and Versatile-Model 3 similar to that in the original network, as shown in Table 2. For example, numbers of filters in the first convolutional layer of the base Model 3 are 20 and 7, respectively. However, their output channels are 20 and 21, respectively, since a spatial versatile filter with size $5 \times 5$ can produce three channels outputs simultaneously.

It can be found in Table 2, Model 3 obtained a higher result ($99.22\%$) with this strategy, which is slightly higher than that of the baseline model. The reason is that if differences between bias terms are extremely large, gradients of secondary convolution filters will be fundamentally different, which makes the training of entire convolution filters difficult. The performance of Model 3 is slightly higher than that of the baseline model, but has significantly lower memory usage and FLOPs, which demonstrates the effectiveness of the proposed versatile convolution filters. In addition, the detailed structure of Versatile-Model 3 in Table 2 and the corresponding demo code for verifying the proposed method can be found in our supplementary materials.

**Filter Visualization:** Convolution filters are used for extracting intrinsic information from natural images. Thus, these filters often present some specific structures, such as line, blob, *etc*. However, the proposed versatile convolution filters adopt a more complex approach to capture useful information from input images, *i.e.*, a large filter consists of a series of smaller filters and each of them will employ on the input images to generate feature maps. Therefore, it is necessary to visualize and compare filters in original CNN and the network using the proposed versatile convolution filters for having an explicit illustration.

Fig. 3 illustrates convolution filters in the first layer of the Baseline and Model 3 in Table 2, respectively. Since the proposed approach is fundamentally different to the original convolution filters, filters using Fcn. 3 present more complex structures. Specifically, each $3 \times 3$ area in Fig. 3 (b) still can be seen as an independent convolution filter with complex structure and obvious magnitude change. In contrast, some $3 \times 3$ areas in Fig. 3 (a) are extremely smooth, which cannot provide distinctive information.

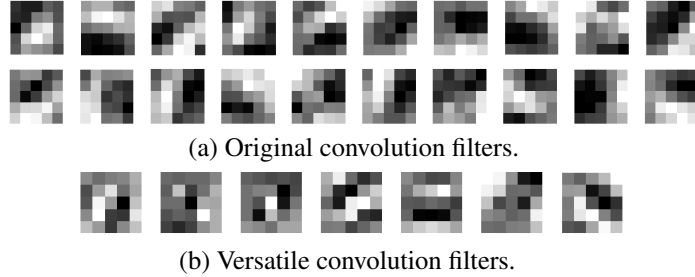

(a) Original convolution filters.

(b) Versatile convolution filters.

Figure 3: Visualization of example filters learned on MNIST.

**Channel versatile filters:** After investigating the effectiveness of the proposed spatial versatile convolution operation, we shall further test the performance of the proposed channel versatile filters as described in Fcn. 8, namely versatile v2. Note that, for the first layer and the last layer in neural networks, we do not apply the channel stride approach, since the input channel of the first layer is usually very small and the output channel of the last layer is exactly the number of ground-truth labels.

There are two important parameters in Fcn. 8, *i.e.*, the number of channels $\hat{c}$ of the convolution filter $\hat{f}$ and the stride $g$. We then established three models using the proposed versatile filter with different $\hat{c}$ and $g$, and trained them on the MNIST dataset as detailed in Table. 3.

Table 3: The performance of the proposed channel versatile filters on MNIST.

| Model | $c - \hat{c}$ | g | Weights | Memory | FLOPs | Accuracy |
|---|---|---|---|---|---|---|
| Baseline | - | - | $4.3 \times 10^5$ | $1681.6KB$ | $22.93 \times 10^5$ | 99.20% |
| Versatile v2-Model 1 | 1 | 1 | $1.18 \times 10^5$ | $460.5KB$ | $12.17 \times 10^5$ | 99.18% |
| Versatile v2-Model 2 | 1 | 2 | $1.18 \times 10^5$ | $460.5KB$ | $11.17 \times 10^5$ | 99.15% |
| Versatile v2-Model 3 | 2 | 1 | $0.79 \times 10^5$ | $309.1KB$ | $12.12 \times 10^5$ | 99.07% |

As mentioned above, the channel versatile filters can reduce the number of convolution filters by a factor of $n = (c - \hat{c})/g + 1$, therefore, when we set $\hat{c} - c = 1$ and $g = 1$, we can reduce about half convolution filters and maintain the similar amount of feature maps. For example, the size of the second layer's convolution filter in Versatile-Model 3 is $5 \times 5 \times 21 \times 17$, and the size of the second layer's convolution filter in Versatile v2-Model 1 is $5 \times 5 \times 21 \times 9$. As a result, the Versatile v2-Model 1 achieved a 99.18 accuracy, which is slightly lower than that of the baseline model, but its memory usage and FLOPs have been reduced significantly.

Similarly, when we set $\hat{c} - c = 1$ and $g = 2$ (*i.e.*, Versatile v2-Model 2), the network obtained similar results to those of Versatile v2-Model 1. Furthermore, when $\hat{c} - c = 2$ and $g = 1$ in Versatile v2-Model 3, the number of convolution filters will be further reduced. However, since the number of filters is very small, the representability of this network is also lower. The Versatile v2-Model 3 with the smallest memory usage and FLOPs obtained a 99.07% classification accuracy. Therefore, we set $\hat{c} - c = 1$ and $g = 1$ in the following experiments for having a best trade-off.

## 3.2 Large Scale Visual Recognition Experiments

Experiments in the above chapter show that the proposed spatial versatile filters in Fcn. 3 and the channel versatile filters are able to replace the traditional convolution operation on the MNIST dataset. We next employed the proposed method on an extremely large image dataset, namely ImageNet ILSVRC 2012 dataset [17], which contains over 1.2M training images and 50k validation images. Three baseline architectures, AlexNet [11], ResNet-50 [7] and ResNeXt-50 [25], were selected for conducting the following experiments. Note that, all training settings such as weight decay and learning rate used the default setting to ensure fair comparisons.

**AlexNet:** AlexNet is one of the most classical deep CNN models for large scale visual recognition, which has over 230$MB$ parameters and a 80.2% accuracy on the ImageNet dataset with 1000 different categories. This network has 8 convolutional layers, sizes of convolution filters in the first six layers

are larger than $1 \times 1$, *i.e.*, $11 \times 11 \times 3 \times 96$, $5 \times 5 \times 48 \times 256$, $3 \times 3 \times 256 \times 384$, $3 \times 3 \times 192 \times 384$, $3 \times 3 \times 192 \times 256$, and $6 \times 6 \times 256 \times 4096$.

Since sizes of convolution filters used in this network are much larger than that in other networks, resources required by this network can be significantly reduced by exploiting the proposed versatile convolution operation. For example, the parameter for the first convolutional layer is $s_1 = \lceil 11/2 \rceil = 6$, thus the number of parameters in this layer with versatile convolution filters is only $11 \times 11 \times 3 \times 16$. In this manner, we established a new network (Versatile-AlexNet in Table 4) and reduced the number of filters in each convolutional layer according to its versatile parameter $s$. Specifically, sizes of convolution filters in its first six convolutional layers are $11 \times 11 \times 3 \times 16$, $5 \times 5 \times 48 \times 86$, $3 \times 3 \times 258 \times 192$, $3 \times 3 \times 192 \times 192$, $3 \times 3 \times 192 \times 128$, and $6 \times 6 \times 256 \times 1366$, respectively. After training the network on the ImageNet dataset, Versatile-AlexNet using Fcn. 3 obtained a $19.5\%$ top5-err and a $42.1\%$ top1-err, which are better than those of the baseline model. The memory usage of filters was reduced by a factor of $1.76\times$, and the FLOPs in Versatile-AlexNet is $1.95\times$ less than that in the baseline model.

Furthermore, we applied the channel versatile filters (Fcn. 8) on the Versatile-AlexNet model with $\hat{c} - c = 1$, and $g = 1$, namely, Versatile v2-AlexNet. In this manner, the number of convolution layer in each layer will be reduced by a factor of $\frac{1}{2}$. As a result, this network achieved a $20.7\%$ top5-err, which is slightly higher than that of the baseline model. But, the memory usage of the entire network is only $73.6MB$, which is only about $30\%$ to that of the baseline model.

Table 4: Statistics for versatile filters on the ImageNet 2012 dataset.

| Model | Weights | Memory | FLOPs | Top1err | Top5err |
|---|---|---|---|---|---|
| AlexNet [11] | $6.1 \times 10^7$ | $232.5MB$ | $0.7 \times 10^9$ | $42.9\%$ | $19.8\%$ |
| Versatile-AlexNet | $3.5 \times 10^7$ | $131.8MB$ | $0.4 \times 10^9$ | $42.1\%$ | $19.5\%$ |
| Versatile v2-AlexNet | $1.9 \times 10^7$ | $73.7MB$ | $0.4 \times 10^9$ | $44.1\%$ | $20.7\%$ |
| ResNet-50 [7] | $2.6 \times 10^7$ | $97.2MB$ | $4.1 \times 10^9$ | $24.7\%$ | $7.8\%$ |
| Versatile-ResNet-50 | $1.9 \times 10^7$ | $75.6MB$ | $3.2 \times 10^9$ | $24.5\%$ | $7.6\%$ |
| Versatile v2-ResNet-50 | $1.1 \times 10^7$ | $41.7MB$ | $3.0 \times 10^9$ | $25.5\%$ | $8.2\%$ |
| ResNeXt-50 [25] | $2.5 \times 10^7$ | $95.3MB$ | $4.2 \times 10^9$ | $22.6\%$ | $6.5\%$ |
| Versatile v2-ResNeXt-50 | $1.3 \times 10^7$ | $50.0MB$ | $4.0 \times 10^9$ | $23.8\%$ | $7.0\%$ |

**ResNets:** To further illustrate the superiority of the proposed scheme, we then employed it on the ResNet-50 model. Although there are many layers with $1 \times 1$ filters in this network, it also has a lot of convolutional layers with large filters, *e.g.* $3 \times 3$ and $7 \times 7$, which accounts for about half memory usage of the entire network. In addition, ResNets introduce shortcut operations which also provides considerable versatile features since receptive fields of neurons in different layer are various as discussed in [18]. Therefore, it is meaningful to investigate the functionality of the versatile convolution filters on this network.

Similarly, we reset the original convolutional layers with the proposed versatile convolution filters. For instance, a convolutional layer of size $3 \times 3 \times 64 \times 128$ will be converted into a new layer of size $3 \times 3 \times 64 \times 32$ using the proposed versatile convolution filters. The performance of the original ResNet-50 and the network using versatile filters were detailed in Table 4.

As mentioned above, there are still considerable filters in the ResNet whose sizes are larger than $1 \times 1$. Thus, its memory usage and FLOPs were reduced obviously by exploiting the proposed versatile convolution filters. Versatile-ResNet-50 with the same amount feature maps achieved a $7.6\%$ top-5 accuracy, which is slightly lower than that of the baseline models with only $75.6MB$ and $3.2 \times 10^9$ FLOPs.

In addition, Versatile v2-ResNet-50 with the same amount feature maps achieved a $8.2\%$ top-5 accuracy, which is slightly higher than that of the baseline models. Its memory usage is only about $41.7MB$, which is only about $\frac{1}{2}$ to that of the original network. Therefore, our Versatile v2-ResNet-50, which is a more portable alternative to the original ResNet-50 model.

Moreover, we attempted to replace original convolution filters by the proposed versatile convolution filters in ResNeXt-50. This network is an enhanced version of ResNet-50, which divides convolutional layers into several smaller groups and achieves a higher performance, and avoids larger convolution

filters. Since more than $90\%$ convolution filters in this network are $1 \times 1$ filters, Fcn. 3 cannot obtain an obvious enhancement. However, the proposed channel versatile scheme in Fcn. 8 can effectively reduce the number of massive $1 \times 1$ convolution filters. Thus, we directly applied the versatile v2 convolution filters with channel stride approach on it. After applying the proposed versatile convolution filter on the ResNeXt-50 model, we obtained a $7.0\%$ top-5 classification error rate, which is slightly higher than its baseline model with only about half memory usage. The detailed statistics of the Versatile v2-ResNeXt-50 using the proposed versatile filters were also shown in Table 4.

Comparing Versatile v2-ResNet-50 and Versatile v2-ResNeXt-50, we found that the memory usage of Versatile v2-ResNet-50 is lower than that of the Versatile v2-ResNeXt-50. This is because the proposed versatile filters can effectively reduce the memory and FLOPs of filters whose sizes are larger than $1 \times 1$, which provides a more flexible way for designing CNNs with high performance and portable architectures.

### 3.3 Comparing with Portable Architectures

Besides sophisticate CNNs such as AlexNet and ResNet-50 with heavy architectures, a variety of recent works attempt to design neural networks with portable architectures and comparable performance. MobileNet [9] utilized separable convolution to reduce memory usage and computational cost of massive large convolution filters. ShuffleNet [26] further proposed shuffle operation to mixed features in different groups and achieved higher results.

Table 5: An overall comparison of state-of-the-art portable CNNs on the ILSVRC2012 dataset.

| Model | Weights | Memory | FLOPs | Top1err |
|---|---|---|---|---|
| 1.0 MobileNet-224 [9] | $0.4 \times 10^7$ | $16.0MB$ | $0.5 \times 10^9$ | $29.4\%$ |
| ShuffleNet $2\times$ (g = 3) [26] | $0.7 \times 10^7$ | $20.6MB$ | $0.5 \times 10^9$ | $26.3\%$ |
| Versatile v2- ShuffleNet $2\times$ (g = 3) | $0.4 \times 10^7$ | $14.0MB$ | $0.5 \times 10^9$ | $27.6\%$ |

Table 5 summarizes state-of-the-art CNN architectures, including their memory usages, FLOPs, and recognition results on the ILSVRC 2012 dataset. Obviously, MobileNet has the smallest model size and FLOPs, but its classification accuracy is lower than those of other networks. ShuffleNet with the similar FLOPs to that of the MobileNet achieves a higher accuracy, with a slightly higher memory usage. By exploiting the proposed versatile convolution filters on the ShuffleNet $2\times$ (g = 3), we reduced more than $30\%$ weights of convolution filters and achieved the smallest model size with a comparable accuracy, which is a more portable convolutional neural network.

In addition, to investigate the generalization ability of the proposed versatile convolution filter, we further employed it on the single image super-resolution experiment. We selected the VDSR (Very Deep CNN for Image Super-resolution [10]) as the baseline model and the Versatile-VDSR with the same amount of feature maps but less memory usage and FLOPs achieved a higher performance. Detailed experiments and analysis can be found in the supplementary materials.

## 4 Conclusions and Discussions

Exploring convolutional neural networks with low memory usage and computational complexity is very essential so that these models can be used on mobile devices. In fact, the main waste in a general neural network is that a convolution filter with massive parameters can only produce one feature for a given data. In order to make full use of convolution filters, this paper proposes versatile convolution filters from spatial and channel perspectives. Thus, we can use fewer parameters to generate the same amount of useful features with a lower computational complexity at the same time. Experiments conducted on benchmark image datasets and models show that the proposed method can not only reduce the requirement of storage and computational resources, but also enhance performance of CNNs, which is very effective for establishing portable CNNs with high accuracies. In addition, the proposed method can be easily implemented using the existing convolution component, we will further embed it into other applications such as object detection and image segmentation.

**Acknowledgments:** This work was supported in part by the ARC DE-180101438, FL-170100117, DP-180103424, and NSFC under Grant 61876007, 61872012. We also thank Huawei Hisilicon for their technical supports.

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
