[Supplementary Material]

# Learning Versatile Filters for Efficient Convolutional Neural Networks (Supplementary Materials)

**Yunhe Wang[1], Chang Xu[2], Chunjing Xu[1], Chao Xu[3], Dacheng Tao[2]**
[1] Huawei Noah's Ark Lab
[2] UBTECH Sydney AI Centre, SIT, FEIT, University of Sydney, Australia
[3] Key Lab of Machine Perception (MOE), Cooperative Medianet Innovation Center,
School of EECS, Peking University, Beijing, China
yunhe.wang@huawei.com, c.xu@sydney.edu.au, xuchunjing@huawei.com
xuchao@cis.pku.edu.cn, dacheng.tao@sydney.edu.au

In this document we give the detailed back-propagation of the proposed versatile convolution filters and summarize the feed-forward and back-propagation procedure in Alg. 1. In addition, we provide the single image super-resolution experiments using original convolution filters and the proposed versatile convolution filters, respectively.

## 1  Back Propagation for Spatial Versatile Convolution Filters

A novel convolution operation was proposed in Section 2.1 in the main body, which extracts $s$-dimensional features in every location in the input data $x$. Thus, the conventional back-propagation scheme should be adjusted accordingly.

The convolution operation in CNNs can be reformulated as the matrix multiplication, and the input data $x$ of a given sample is usually divided into several overlapping areas in practice. Dividing the input data $x$ into $k = H' \times W'$ areas (the size of each area is $d \times d \times c$) and vectorizing them, we have $X = [\text{vec}(x_1), ..., \text{vec}(x_k)] \in \mathbb{R}^{d^2 c \times k}$. Similarly, we reformulate the output feature map $y$ and its corresponding convolution filter $f$ as $Y = \text{vec}(y) \in \mathbb{R}^{k \times 1}$ and $F = \text{vec}(f) \in \mathbb{R}^{d^2 c \times 1}$, respectively. Thus, the conventional convolution (Fcn. 1 in the main body) can be rewritten as:

$$Y = X^\top F + b. \tag{1}$$

Denote the gradient for the output data $Y$ is $\partial \mathcal{L} / \partial Y$, gradients of $X$ and $F$ can be calculated according to the back-propagation strategy:

$$\frac{\partial \mathcal{L}}{\partial X} = F \left( \frac{\partial \mathcal{L}}{\partial Y} \right)^\top, \quad \frac{\partial \mathcal{L}}{\partial F} = X \left( \frac{\partial \mathcal{L}}{\partial Y} \right), \tag{2}$$

where $\mathcal{L}$ is the loss function of the entire neural network. Similarly, the matrix form of the proposed versatile convolution is

$$Y = [X^\top (F_1) + b_1, ..., X^\top (F_s) + b_s], \\ s.t. \quad s = \lceil d_i / 2 \rceil, \quad F_i = \text{vec}(M_i \circ f), \tag{3}$$

where $Y = [Y_1, ..., Y_s] \in \mathbb{R}^{k \times s}$ is the output consists of versatile feature maps. We can regard each sub feature map in $Y$ as the output of an individual convolution filter, gradients of input data and convolution filters are

$$\frac{\partial \mathcal{L}}{\partial X} = \sum_{i=1}^{s} \left[ F_i \left( \frac{\partial \mathcal{L}}{\partial Y_i} \right)^\top \right], \quad \frac{\partial \mathcal{L}}{\partial F_i} = X \left( \frac{\partial \mathcal{L}}{\partial Y_i} \right). \tag{4}$$

There are several zeros in $F_i$ when $i > 1$, and weights corresponding to these positions in the entire $F$ do not participate the calculation in Fcn. 3, but the above function will generate a dense vector

**Algorithm 1** Feed-Forward and Back-Propagation of the Spaital Versatile Convolution Filter.

---

**Input:** A convolutional layer with $n$ convolution filters $[f_1, .., f_n]$, bias terms $[b_1, .., b_n]$, input data $x$, the loss function $\mathcal{L}$, and the learning rate $\eta$.

1: Calculate $s = \lceil d_i/2 \rceil$, and generate $s$ masks $\{M\}_{i=1}^{s}$ according to Fcn. 3 in the main body;
2: Convert $x$ into a matrix: $X \leftarrow [\text{vec}(x_1), ..., \text{vec}(x_k)]$;
3: **Feed Foward:**
4: **for** $i = 1$ to $s$ **do**
5:     $F_i \leftarrow [\text{vec}(M_i \circ f_1), ..., \text{vec}(M_i \circ f_n)]$;
6:     Obtain feature maps $Y_i \leftarrow X^\top F_i + b_i$;
7: **end for**
8: **Back Propagation:**
9: **for** $i = 1$ to $s$ **do**
10:     Calculate the gradient of feature maps $\partial\mathcal{L}/\partial Y_s$;
11:     $\partial\mathcal{L}/\partial X_i \leftarrow F_i \left( \frac{\partial\mathcal{L}}{\partial Y_i} \right)^\top, \partial\mathcal{L}/\partial F_i \leftarrow X \left( \frac{\partial\mathcal{L}}{\partial Y_i} \right)$;
12: **end for**
13: Aggregate gradients of $X$ and $F$:
    $\partial\mathcal{L}/\partial X = \frac{1}{s} \sum_{i=1}^{s} \partial\mathcal{L}/\partial X_i, \quad \partial\mathcal{L}/\partial F = \frac{1}{s} \sum_{i=1}^{s} [\partial\mathcal{L}/\partial F_i \circ \text{vec}(M_i)]$;
14: Calculate new filters $\hat{F} \leftarrow F - \eta\partial\mathcal{L}/\partial F$, and update $[f_1, ..., f_n]$ accordingly;

**Output:** Feature maps $Y \leftarrow [Y_1, ..., Y_m]$ and filters $F$.

---

$\partial\mathcal{L}/\partial F_i$. Thus gradients of these weights should be discarded. The gradient of the entire convolution filter is

$$\frac{\partial\mathcal{L}}{\partial F} = \sum_{i=1}^{s} \left[ \frac{\partial\mathcal{L}}{\partial F_i} \circ \text{vec}(M_i) \right]. \tag{5}$$

In addition, both convolution filters $F$ and input data $X$ are utilized multiple times in the proposed method, and their gradients are aggregated from different scales, which is $s\times$ larger than those in the conventional CNNs. Therefore, we divide them by the number of scales $s$ to avoid the gradient explosion in practice, and filters will be updated according to the learning rate $\eta$, *i.e.* $F = F - \partial\mathcal{L}/\partial F$. Alg. 1 provides the detailed back-propagation procedure of the proposed versatile convolution filter.

## 2 Image Super-resolution Experiments

The superiority of the proposed method was demonstrated in the main body of this paper under the visual recognition experiments. Actually, convolution neural networks can be utilized to solve a large variety of real-world applications such as image denoising, visual segmentation, and image style translation. To evaluate the generalization ability of the proposed versatile convolution operation, we then applied it on the single image super-resolution problem. The image super-resolution task receives a low-resolution image and then outputs its high-resolution estimation. Different from conventional CNNs for visual classification and detection which exploits convolution filters for extracting powerful features, filters in these models are employed for making output images clear and visually pleasant. Therefore, it is very meaningful to investigate the performance of the proposed versatile convolution filters on this task.

We selected VDSR (Very Deep CNN for Image Super-resolution [2]) as the baseline model for conducting the image super-resolution task. The baseline model contains 22 convolutional layers with a number of $3 \times 3$ convolution filters, which trained on a benchmark dataset consists of 291 images. Each image in this dataset is first divided into several patches and then augmented with some commonly used strategies (*i.e.* rotation and flip) to form the training set. Although the VDSR model utilizes a relatively small dataset, but shows better performance than that of SRCNN [1] trained on the ILSVRC dataset due to it contains more covolutional layers.

Similar to experiments in the main body, a new model using the proposed spatial versatile filters and another model using the proposed channel versatile filters were established, respectively. Then, the baseline VDSR model and the two new models were trained on the dataset using the same setting (*e.g.* learning rate, number of epochs) used in [2], respectively. Images in the dataset were downscaled by $2\times$ and $4\times$ in order to train models for processing images with different resolutions.

| Ground-truth | VDSR | Versatile-VDSR | Versatile v2-VDSR |
|---|---|---|---|
| *Baby* (×4) | PSNR = 33.40*dB* | PSNR = 33.45*dB* | PSNR = 33.43*dB* |
| *Butterfly* (×2) | PSNR = 34.45*dB* | PSNR = 34.61*dB* | PSNR = 34.47*dB* |

Figure 1: Image super-resolution results of the baseline VDSR model and the proposed versatile filters, where the top line are results of the *Baby* (×4) image, and the bottom line are results of the *Butterfly* (×2) image.

Detailed results are shown in Table 1. PSNR values were calculated by comparing output images and ground-truth high-resolution images and FLOPs were calculated using $256 \times 256$ images.

Table 1: Statistics for versatile filters on VDSR.

| Model | Memory | FLOPs | PSNR (×2) | PSNR (×4) |
|---|---|---|---|---|
| VDSR [2] | 2.82*MB* | $48.39 \times 10^9$ | 37.53*dB* | 31.35 *dB* |
| Versatile-VDSR | 1.41*MB* | $26.90 \times 10^9$ | 37.64*dB* | 31.41 *dB* |
| Versatile v2-VDSR | 0.69*MB* | $26.46 \times 10^9$ | 37.58*dB* | 31.37 *dB* |

It can be found in Table 1 that, memory usage and computational complexity of networks using the proposed versatile convolution operation have been reduced significantly, and PSNR values of Versatile-VDSR with the same amount of feature maps are higher than those of the baseline model, while the memory usage of this model is only about $1.41MB$. Compared with the baseline model, MS-VDSR achieved a $1.99\times$ compression ratio and a $1.79\times$ speed-up ratio. In addition, the memory usage of the Versatile v2-VDSR model is only about $1.41MB$ with the similar performance to that of the original model.

Fig. 1 illustrates some visualization results of the baseline model and the proposed method. Results generated by networks using the proposed versatile convolution filters are better than those of the baseline model since we can provide the same amount feature maps with multi-scale information, which is able to make the estimation smooth in every scale.