[Reviews · NeurIPS 2018]

Reviewer 1



The paper introduces two new types of convolutional filters, named versatile filters, which can reduce the memory and FLOP requirements of conv. nets. The method is simple and, according to the experimental results, it seems to be effective. The text quality is OK, although it would definitively benefit from better explanations about the proposed method. For instance, Fig. 1 is a bit confusing (Are you showing 4 different filters in Fig 1 (b) ?). My main concerns about this paper are related to the experiments and results, as detailed in the following questions: (1) Regarding the FLOP reduction, it is not clear how the reduction in the number of computations is really achieved. In the algorithm presented in the supplementary material, I cannot see any reduction in the number of FLOPs. Although you have less filters, these filters must be applied s times, and you still added the overhead of the masks. Please explain how you can achieve the FLOP reduction in practice. (2) The memory and FLOP reductions reported in the experiments are just nominal, correct? You must also report the actual memory and execution time of each one of the methods. For instance, ShuffleNet paper reports the actual inference time for different variations of their method as well as the baselines. Without this info, it is difficult to judge the real effect of the proposed method. (3) How did you implement the proposed conv. nets for the ImageNet experiment? There is no information about the implementation in the paper. I see you used Matlab for the MNIST experiment. How easy is it to implement the prosed method using one of the mainstream DL libraries (TensorFlow, Pytorch, Caffe) and have the proposed FLOP reduction? --- Given the author feedback, I am willing to increase my score from 5 to 6. The author feedback properly addressed just some of my questions. Fig.1 is still unclear to me and the execution time comparison with at least one previous model (ShuffleNet) is not reported.

Reviewer 2



The paper focuses on developing CNN that is usable at mobile devices. The authors believe that the main problem of massive parameters comes from the convolutional filter, and they introduce spatial- and channel-based versatile filters. Experiments are also well designed, results (especially Table 4) verify the claims raised by the authors. I think it is a neat article, which is acceptable.

Reviewer 3



Convolutional neural networks (CNNs) form one of the most important family of deep neural networks and have provided powerful applications in various domains including image classification and speech recognition. There has been a strong remand in reducing further the computational complexity of deep CNNs and some methods have been proposed in the literature. The purpose of this paper is to propose a new method of versatile filters by treating filters in two levels where the secondary filters are derived from the primary ones without using more storage. This method can improve the capacity of the CNN filters. Two classes of versatile filters are introduced from a spatial or channel perspective. Experiments are also carried out to illustrate the method. The method is promising and the presentation of the paper is nice.